# Coffee as an Identifier of Inflation in Selected US Agglomerations

**Marek Vochozka \***, **Svatopluk Janek and Zuzana Rowland**

Institute of Technology and Business in Ceske Budejovice, Okružní 517/10,
370 01 České Budějovice, Czech Republic
\* Correspondence: vochozka@mail.vstecb.cz

**Abstract:** The research goal presented in this paper was to determine the strength of the relationship between the price of coffee traded on ICE Futures US and Consumer Price Indices in the major urban agglomerations of the United States—New York, Chicago, and Los Angeles—and to predict the future development. The results obtained using the Pearson correlation coefficient confirmed a very close direct correlation (r = 0.61 for New York and Chicago; r = 0.57 for Los Angeles) between the price of coffee and inflation. The prediction made using the SARIMA model disrupted the mutual correlation. The price of coffee is likely to anchor at a new level where it will fluctuate; on the other hand, the CPIs showed strong unilateral pro-growth trends. The results could be beneficial for the analysis and creation of policies and further analyses of market structures at the technical level.

**Keywords:** coffee price; CPI; time series; Pearson r; SARIMA model

## 1. Introduction

After oil, coffee is one of the most traded commodities in the world commodity markets [1]. Its consumption is nearly a ritual, and it is consumed on a regular basis because of its stimulating effects [2]. On the other hand, it shall be noted that its excessive consumption may negatively affect public health [3]. It is even the processing of coffee beans that is unhealthy for human organisms, as according to numerous studies, coffee residues are toxic [4]. Lopez-Garcia et al. [5] are skeptical in terms of the negative long-term impact on habitual coffee consumers' health, claiming that, based on their observations, adverse effects cannot be confirmed. It can thus be stated that mainly the excessive consumption of coffee has some effects on human health, which are not always negative. Empirical results are varied.

From a macroeconomic perspective, commodity markets are closest to the theoretical concept of a perfect market, i.e., an environment where the producer´s marginal costs are constant [6]. Nevertheless, the major coffee producers have a significantly better negotiating position, which is a typical feature of a monopoly. This means they have bargaining power that is able to influence the prices of commodities listed in the commodity markets. However, it shall be specified that within the supply-customer chain, this position belongs to intermediaries rather than producers [7]. This is also confirmed by Usman and Callo-Concha [8], who stated that, for example, coffee producers in Ethiopia have very difficult access to the market. From the point of view of the supply power, coffee as a crop is influenced by natural factors such as pests and climate [9].

In contrast, smaller producers from developing countries are more dependent on the demand. With lower prices, they are forced to dislocate the working capital, which has a negative causal effect on the social and health conditions of the inhabitants [10].

Based on the above, it can be concluded that commodity coffee markets are strongly influenced by shocks both on the supply and demand side [11]. The findings of many studies suggest that coffee shows an interesting correlation with other economic variables.

For example, Salisu et al. [12] stated that predicting coffee prices is of great importance because the price of coffee directly correlates with the level of inflation, especially in large urban agglomerations in the United States. On the other hand, Creti et al. [13] described the correlation between the stock and commodity markets, pointing out the volatility in the market after 2008, i.e., after the start of the global economic crisis. The importance of predicting future development was also emphasized by Vochozka et al. [14] using the example of copper, whose price increased dramatically as a result of the COVID-19 pandemic. Similarly, Vochozka et al. [15] mentioned the importance of predicting the development of Brent oil prices as one of the most important commodities.

From an economic point of view, it seems attractive to find a simple economic quantity that could, based on its development, define the development of an important macroeconomic indicator, such as the level of inflation. Coffee, which is among the most important economically traded commodities, is also a necessary and inferior good [16]. It is the necessity of coffee as an economic good that determines it as a suitable input.

The goal of this paper is to determine the relationship between the coffee price in the commodity stock market ICE Futures US and the level of inflation in selected US agglomerations (New York, Chicago, Los Angeles) as well as to forecast the future development of both economic variables. For these purposes, two research questions were formulated as follows:

RQ1: How strong is the correlation between the coffee price in ICE Futures US and the level of inflation in selected US agglomerations?

This RQ was formulated on the basis of premises indicating a direct correlation between coffee price and inflation with the aim to identify the strength of the correlation and thus determine the significance of the relationship. Coffee can thus be an important lead in determining the level of inflation.

RQ2: What trend can be expected for the price of coffee and inflation?

The prediction of the price of coffee and the level of inflation has significant predictive power. The assumption is that the price of coffee will copy a strong upward trend of inflation in the United States and, thus, also in large cities. The price of coffee in the commodity markets is thus expected to grow with rising consumer prices.

## 2. Literature Review

Commodity markets exert a centralizing pressure on commodity trade [17]. Trade in commodity markets entails the risk of loss. This can undermine the confidence of investors, brokers, and social institutions [18]. The fluctuation of prices even affects the balance of banks and overall financial stability [19]. Roch [20] used the heterogeneous panel SVAR model (Structural Vector Autoregression) and stated that a flexible exchange rate, inflation targeting, and clear fiscal policy of the state are necessary for isolating the negative impact of price fluctuations. The SVAR model was also used by Aliyev and Kocenda [21] in the context of the monetary policy of the European Central Bank (ECB, Frankfurt, Germany). Even in this instance, the very significant effect of monetary policy on the prices of food commodities was confirmed.

Endogenous commodity shocks can have a pathogenic effect on the state of the national economy. As an example, the so-called Dutch disease can be mentioned where appreciation of the national currency occurred on the basis of new natural gas deposits in the North Sea, which led to the loss of competitiveness of secondary production [22]. Poncela et al. [23] dealt with the issue of the "Dutch disease" in the specific Colombian economy, which was highly dependent on the export of coffee using vector error correction (VECM). Their conclusions confirmed the direct positive correlation between the exchange rate and the price of coffee as a commodity.

The value of the exchange rate indicates the level of inflation. After the outbreak of the crisis in 2008, the prices of commodities in commodity markets were very high, which was followed by considerable fluctuations. The ARMA–EGARCH (Autoregressive Moving Average–Generalized Autoregressive Conditional Heteroskedasticity) model classifies the

fluctuations of food commodity prices in the long post-crisis period as above the level of equilibrium as a result of monetary policy impulses [24]. Halka and Kotlowski [25] believe that the shocks in the commodity markets influence the level of inflation. This finding was provided by the SVAR model in small and open economies of European countries. A similar conclusion was also made by Forbes [26], who considers commodity prices to be one of the global factors directly affecting the CPI. The NiGEM model provided a similar finding. According to Metelli and Natoli [27], the decrease in global commodity prices has the potential to lower inflation under the assumption of low state investments and zero responses of the central bank. In contrast, in China, Mao et al. [28] showed a negative correlation between the level of inflation and price bubbles in commodity markets, especially in the case of crops such as corn and soybeans. The authors based their assumption on the application of a logistic model.

Classical (conventional) instruments of correlation analysis include the Pearson correlation coefficient and Kendall´s tau [29]. Sajnog and Rogozinska-Pawelczyk [30] used the Pearson correlation coefficient to identify the relationship between profitability ratios and the rewards of managers. The application of the Pearson coefficient also confirmed the existing relationship between the balanced budget of small municipalities and subsidies [31]. On the other hand, C. S. H. Wang et al. [32] noted the divergence of results in testing the informative value of conventional indicators. Volsi et al. [33] used the Pearson correlation coefficient as a tool to determine the production of coffee in Brazil; similarly, Aparecido et al. [34] used the Pearson correlation coefficient to analyze the relationship between the climatic conditions and the ripening time of coffee beans. Revenues from coffee plantations, which would indicate the future transition to efficient agriculture, were also addressed by Martello et al. [35], considering the Pearson coefficient as a reliable indicator. On the other hand, Hofert and Koike [36] view the main shortcoming of the Pearson correlation coefficient to be its inability to capture random variables. Furman and Zitikis [37] recognized the importance of the Gini indicator in economics, insurance, and finance. Compared to the Pearson correlation coefficient, Gini is capable of data modeling with heavy-tailed distributions.

Stehel et al. [38] forecasted the future development of the financial health of agricultural companies using Kohonen networks. Artificial neural networks (ANN) have been used for the analysis of long time series. Thanks to their ability to learn, it is possible to achieve very accurate results, as confirmed by Vrbka et al. [39]. Similar findings have also been achieved by Brabenec et al. [40], who focused on predicting the development of gold prices. Drachal [41] applied the ARIMA model (Autoregressive Integrated Moving Average) for analyzing energy commodities. The predictions provided by this model are considered to be very precise. This was also confirmed by Ouyang et al. [42], who used the ARIMA model as one of the classical predictive models that are often used.

The first research question will be answered based on the application of the Pearson correlation coefficient. The Pearson correlation coefficient remains a widely applicable conventional statistical tool. The predictive model ARIMA, or its seasonal modification SARIMA, which has the potential of providing quality predictions, is classified similarly. Its application for answering the second research question is thus relevant.

## 3. Data and Methods

### 3.1. Data

The data for the analysis of the CPIs were obtained from the US Bureau of Labor Statistics. Specifically, these were the monthly values of the CPIs from large US agglomerations—New York (NY), Chicago (CH), and Los Angeles (LA). The time series included the period from January 2000 to September 2022. Seasonality was not removed from the indices.

The price of coffee was expressed using the data from the portal Yahoo Finance. These were the adjusted closing prices of coffee marked (KC = F) traded on the commodity exchange ICE Futures US. The values are given in US dollar cents per pound (0.453 kg) in monthly data. For the purposes of this paper, the values in cents were converted to

US dollars (USD). As in the case of the CPIs, the time series included the period from January 2000 to September 2022.

*3.2. Methods*

The first research question will be answered on the basis of the application of the Pearson correlation coefficient.

The results of the Pearson correlation coefficient enable the determination of the correlation strength where the resulting r reaches a value in the interval of <−1, 1>. The interval is further specified in Table 1.

**Table 1.** Strength of dependence in intervals.

| Direct dependence (extreme) | r = 1 |
| --- | --- |
| Strong direct dependence | r = <0.5; 0.99> |
| Moderate direct dependence | r = 0.5 |
| Weak direct dependence | r = <0.1; 0.5> |
| No dependence | r = 0 |
| Weak indirect dependence | r = <−0.1; −0.5> |
| Moderate indirect dependence | r = −0.5 |
| Strong indirect dependence | r = <−0.5; −0.99> |
| Indirect dependence (extreme) | r = −1 |

Source: Own processing.

The Pearson correlation coefficient was constructed in RStudio using the statistical language R. The assumption of a direct relationship between the level of inflation and coffee price was confirmed ex ante by means of the identification of the *p*-value. The level of the significance value $\alpha$ to reject the hypothesis $H_0$ was set to 0.05. The alternative hypothesis $H_1$ confirms the existence of a statistically significant correlation between inflation and coffee price.

The second research question will be answered based on the application of the SARIMA model. This model was constructed in the application RStudio using the statistical language R. The prediction was made for all CPIs for the US agglomerations and coffee prices on the commodity exchange ICE Futures. The SARIMA model corresponds to the ARIMA model; the difference consists of the fact that SARIMA considers seasonal fluctuations. The prediction of all monitored economic variables was made for one year, starting from October 2022 to September 2023.

When analyzing in more detail, the models can be specified as ARIMA(p, d, q) and SARIMA(p, d, q)(P, D, Q)m, where p represents the autoregressive component of the model (the number of time lags), d is the degree of differentiation required by the time series, and q is the order of moving average. SARIMA(p, d, q)(P, D, Q)m is a product of the multiplication of non-seasonal factors (p, d, q) by seasonal factors (P, D, Q) where m is the data frequency. The non-seasonal ARIMA(p, d, q) model can be mathematically expressed as follows Clarke and Clarke [43]:

$$\Delta_{y_t}^d = \phi_1 \Delta_{y_{t-1}}^d + \cdots \phi_1 \Delta_{y_{t-p}}^d + \gamma_1 \varepsilon_{t-1} + \gamma_1 \varepsilon_{t-q} + \varepsilon_t \,, \tag{1}$$

where $\Delta_d$ is the differential operator, $y_t$ predicted values, $\phi$ is the coefficient for each parameter *p* and *q*, $\gamma$ is the coefficient for each parameter *q*, and $\varepsilon_t$ is the errors residual at time *t*.

The seasonal SARIMA(p, d, q)(P, D, Q) model can be expressed as follows Clarke and Clarke [43]:

$$\phi_p(B)\Phi_p(B^s)(1-B)^d(1-B^s)^D Y_t = \theta_q(B)\Theta_q(B^s)\varepsilon_t. \tag{2}$$

Individual members can be defined as follows:

$$\phi_p(B) = \left(1 - \phi_1 B - \cdots \phi_p B^p\right), \tag{3}$$

$$\theta_q(B) = \left(1 - \theta_1 B - \cdots \theta_1 B^q\right), \tag{4}$$

$$\Phi_P(B)^s = \left(1 - \Phi_1 B^s - \cdots \Phi_P B^{sP}\right), \tag{5}$$

$$\Theta_Q(B^s) = \left(1 - \Theta_1 B^s - \cdots \Theta_Q B^{sQ}\right). \tag{6}$$

It shall be mentioned that the individual components (p, d, q)(P, D, Q) and the lags of the model in the automatic version are selected independently, or the model itself selects the most optimal variation. AutoSARIMA autonomously tests the best-fit correlations between time series and predictions. The relevance of the prediction model is verified by tests with manually adjusted versions of ARIMA.

## 4. Results

In the long run, the inflation trend is a generally known fact. Figure 1 shows the pro-growth trend of the CPIs in all US agglomerations.

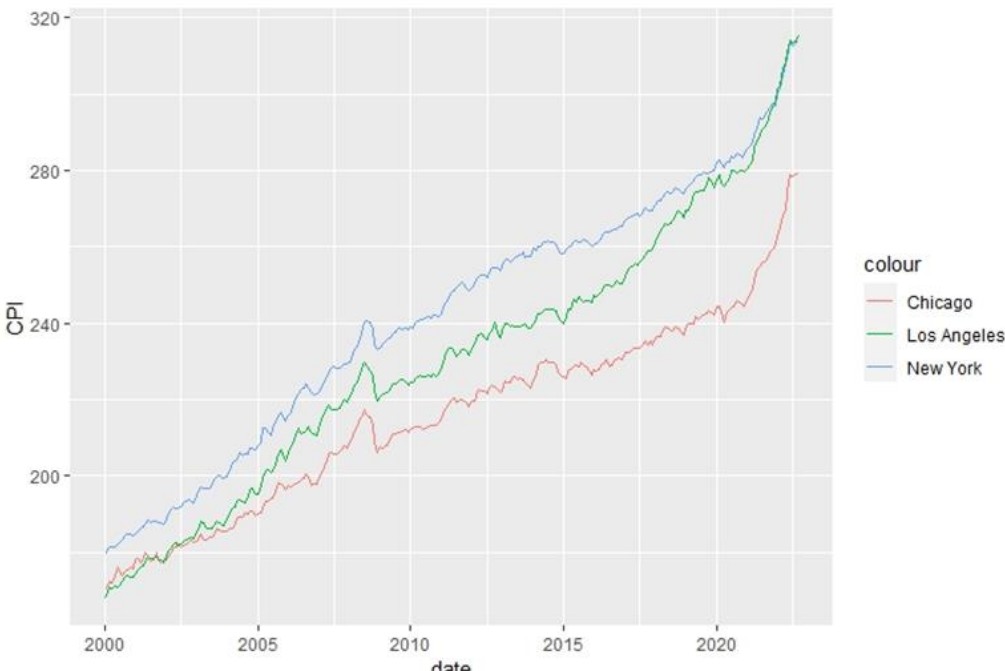

**Figure 1.** Course of CPI in the agglomerations of New York, Chicago, and Los Angeles. Own processing based on data from US Bureau of Labor Statistics.

The highest level of inflation was recorded in New York during the majority of the monitored period. At the beginning of the time series, the second highest level of inflation was recorded in Chicago; however, in the first quarter of 2022, the inflationary pressure moved the LA agglomeration to the second position. In Chicago, the trend of inflation can be generally characterized as pro-growth when compared to the two indices; on the other hand, the growth was not as sharp as in the case of the indices.

What is interesting is the analogous inflationary shock recorded in the case of all monitored indices in the third quarter of 2008 during the outbreak of the mortgage crisis, which resulted in the global economic crisis. This jump was followed by a correction, during which a deflationary trend dominated. The low interest rates set by the FED in order to recover the economy were not the cause of the sharply accelerating inflation since the further growth that followed the deflationary correction was nearly linear.

After 2020, the Los Angeles CPI nearly copied the line of the New York CPI. However, from December 21 to the end of the monitored period, the inflation rate in Los Angeles was the highest, which was an unprecedented situation within the monitored period.

In 2020, there was evidence of the most significant inflationary shock with the outbreak of the global COVID-19 pandemic, which caused numerous cracks in the supply-customer chain due to closures in many industries. Shortages coupled with demand boosted by the low FED interest rates formed a strong basis for potential stagflation in the US economy.

Figure 2 shows the price trend of coffee traded in ICE Futures US. As already known, the price strongly correlates with the state of the world´s coffee reserves (i.e., supply), with the influence of many natural phenomena, such as climate and pests.

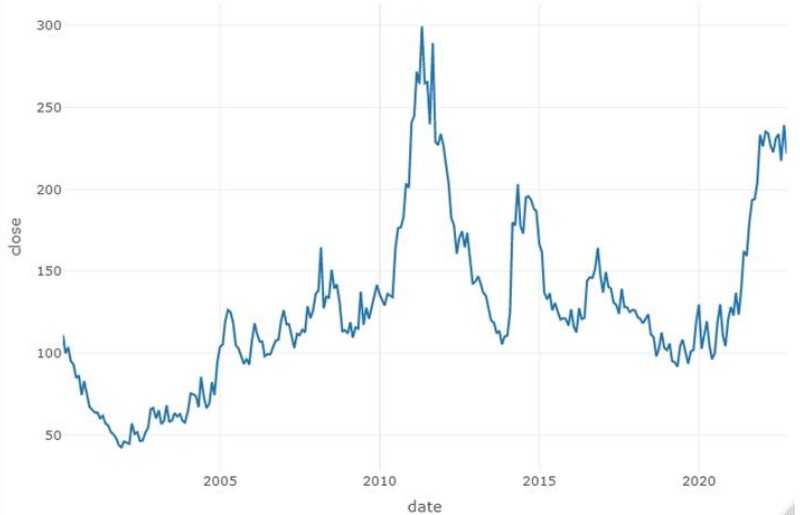

**Figure 2.** Trend of coffee price in ICE Futures US. Own processing based on data from Yahoo Finance.

In the first year of the monitored period, coffee prices fell to their historical minimum in November 2001 (USD0.426). The trend in the following months can be described as volatile (volatility is characteristic of the prices within the whole monitored period) and pro-growth. At the beginning of the Q2 of 2010, there were significant shocks in coffee prices, which culminated in the maximum price for the whole period in April 2011 (USD2.99). This phenomenon occurred in the context of an ongoing economic crisis. After several fiscal and monetary stimuli, extreme price falls could be seen, with prices reaching the minimum in October 2023. This was followed by a second price shock, with its minimum in April 2014 (USD2.03). The following trend can be described as highly volatile and downward. The third shock was recorded in May of the pandemic year 2021, and even after reaching the peak, the prices have not shown any signs of correction or decrease. The market has defined, and, in the future horizon will probably define, the new coffee price level.

### 4.1. Application of the Pearson Correlation Coefficient for Determining the Correlation between Inflation and Coffee Price

The Pearson correlation coefficient was successively applied to all CPIs and coffee prices. The results are presented in Table 2.

**Table 2.** Values of Pearson correlation coefficient and *p*-value.

|  | r | *p*-Value |
|---|---|---|
| Pearson CPI(NY)/Coffee | 0.61 | $p < 2.2 \times 10^{-16}$ |
| Pearson CPI(CH)/Coffee | 0.61 | $p < 2.2 \times 10^{-16}$ |
| Pearson CPI(LA)/Coffee | 0.57 | $p < 2.2 \times 10^{-16}$ |

Source: Own processing.

Figure 3 shows the results of the linear regression and the Pearson correlation coefficient between the New York CPI and coffee prices.

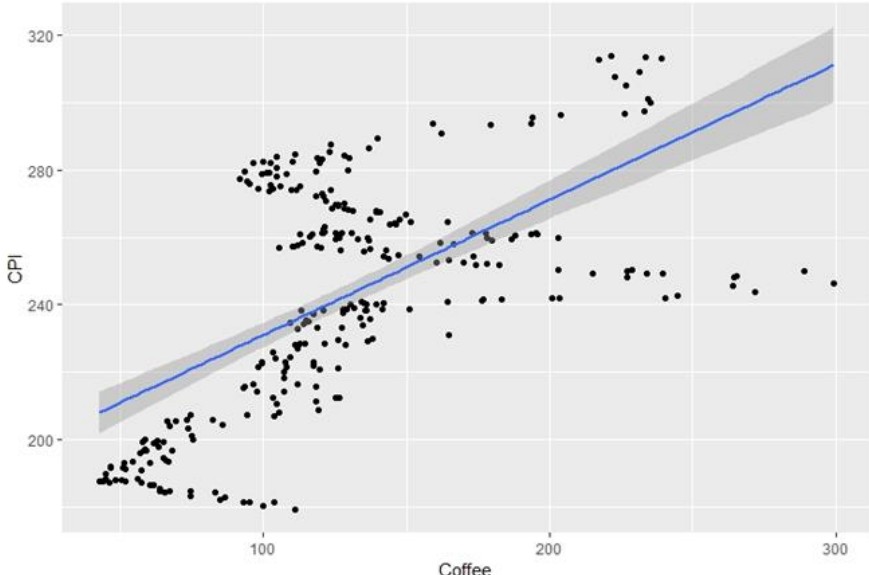

**Figure 3.** Linear regression based on Pearson correlation coefficient between CPI New York and Coffee. Own processing based on data from US Bureau of Labor Statistics and Yahoo Finance.

The extremely low *p*-value accentuates the significant correlation between the level of inflation and coffee price in New York, which means that $H_0$ can be rejected. The relationship between these two variables shows a strong direct dependence with a medium dependence within the interval, as evidenced by the resulting *r* value of 0.61.

Figure 4 shows the correlation between the Chicago CPI and coffee prices. The *p*-value is the same as in the case of New York. It is thus possible to reject $H_0$ and thus confirm the existence of the correlation in Chicago, which is analogous to New York.

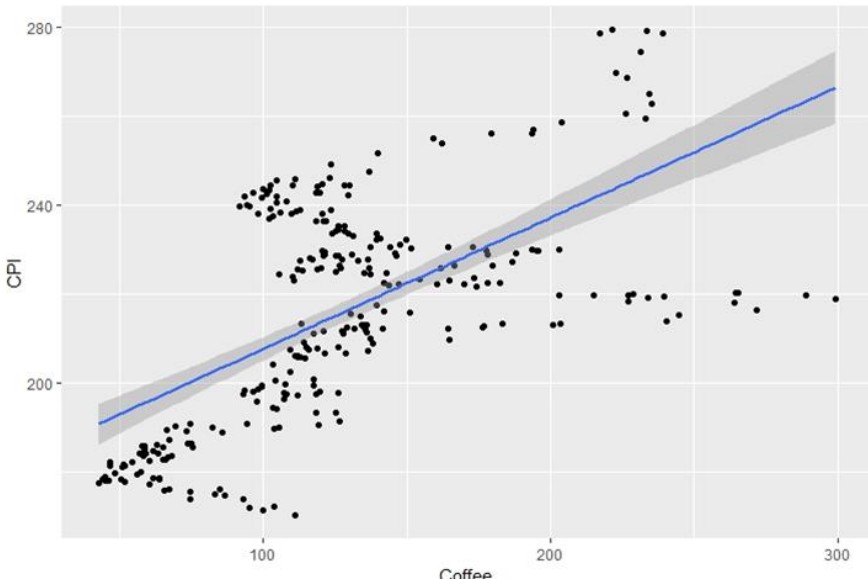

**Figure 4.** Linear regression based on Pearson correlation coefficient between CPI Chicago and Coffee. Own processing based on data from US Bureau of Labor Statistics and Yahoo Finance.

The last correlation analysis was performed for Los Angeles (see Figure 5). The same *p*-value as in the case of the above CPIs indicates the rejection of $H_0$. However, the resulting value of the Pearson correlation coefficient was slightly lower (r = 0.57).

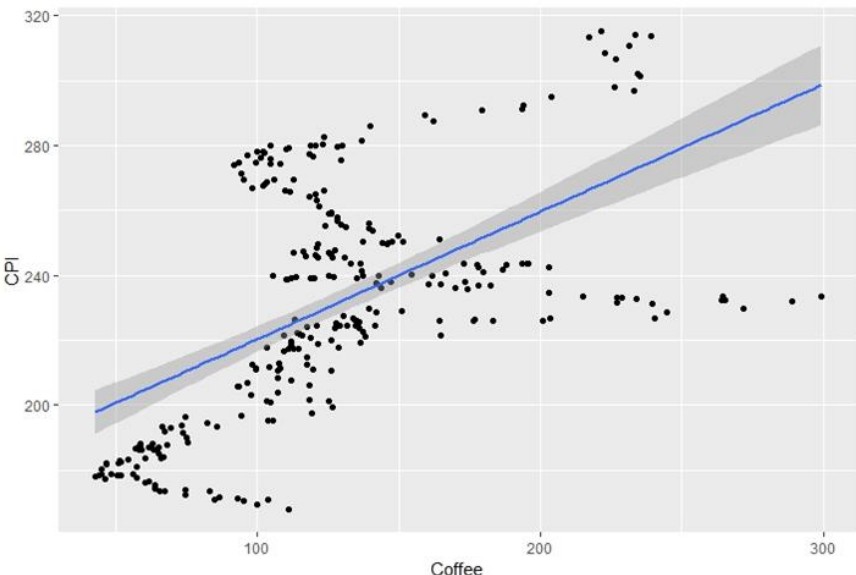

**Figure 5.** Linear regression based on Pearson correlation coefficient between CPI Los Angeles and Coffee. Own processing based on data from US Bureau of Labor Statistics and Yahoo Finance.

This indicates a less close relationship between inflation and coffee price in Los Angeles, with a trend of moderate dependence.

*4.2. Application of the SARIMA Model for Forecasting Coffee Price and Inflation*

Figure 6 shows a 12-month prediction of the coffee price trend (October 2022–September 2023). For illustration, Table A1 with point prediction values is presented in Appendix A.

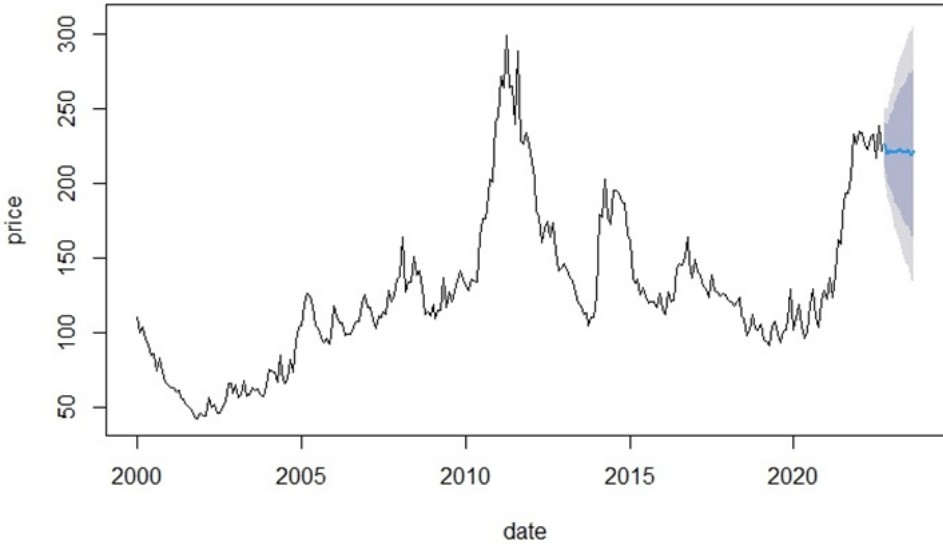

**Figure 6.** SARIMA(2,1,1)(0,0,1) prediction of coffee price trend on the ICE Futures US commodity exchange. Own processing based on data from Yahoo Finance.

According to the AutoSARIMA model, the end of 2022, or rather the whole predicted period, will be characterized by a newly set coffee price level, with weak fluctuations where the prices will range between 2.19 USD/pound and 2.26 USD/pound and neither a pro-growth nor a downward trend could be identified. The most efficient model appeared to be the SARIMA(2, 1, 1)(0, 0, 1) model.

The predicted trend of CPI in New York is described in Figure 7. The individual values of point predictions are presented in Table A2 in Appendix A.

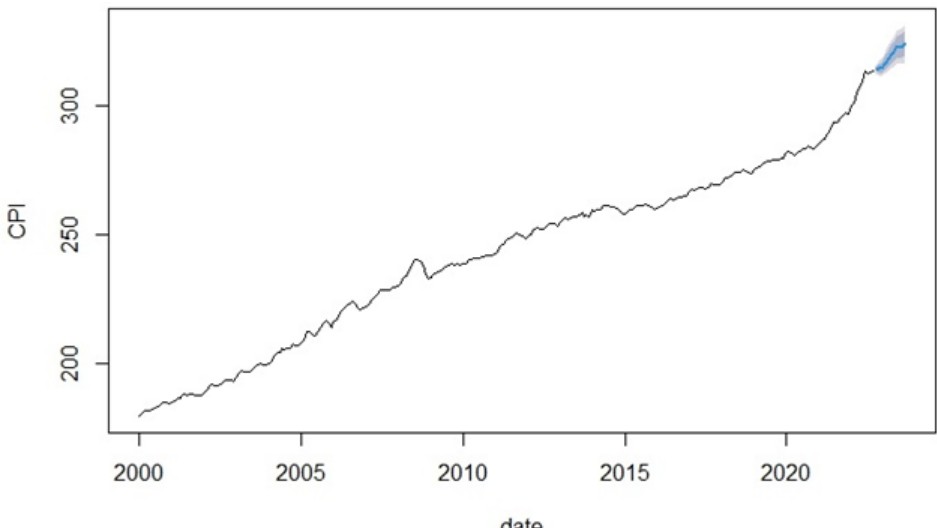

**Figure 7.** SARIMA(1,1,0)(2,0,0) prediction of CPI trend in New York. Own processing based on data from US Bureau of Labor Statistics.

Based on the results of the SARIMA model prediction, the growth of consumer prices can be expected in New York. In October 2023, the last month of the monitored period predicted, the difference between the situation at the beginning of the period (October 2022) was more than USD10. SARIMA(1, 1, 0)(2, 0, 0) was thus evaluated as the most suitable predictive model.

Figure 8 shows the trend of CPI in Chicago. As in the previous cases, the individual values of the prediction are presented in more detail in Appendix A (see Table A3).

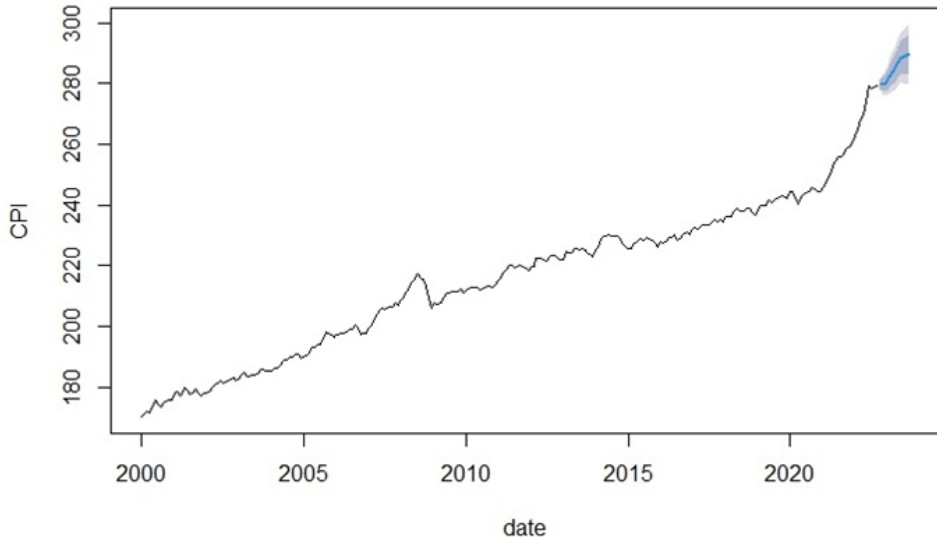

**Figure 8.** SARIMA(1,1,1)(2,0,0) prediction of CPI trend in Chicago. Own processing based on data from US Bureau of Labor Statistics.

As in New York, the predicted inflation trend in Chicago is strongly pro-growth. CPI Chicago shows, quite similarly to CPI New York, the difference between the situation at the beginning and the end of the predicted period was + USD10 in the consumer basket in Chicago (see Table A3). SARIMA(1, 1, 1)(2, 0, 0) was selected as the optimal model.

Finally, the SARIMA model was applied to determine the inflation rate in Los Angeles. The trend is shown in Figure 9; individual values are presented in Table A4 in Appendix A.

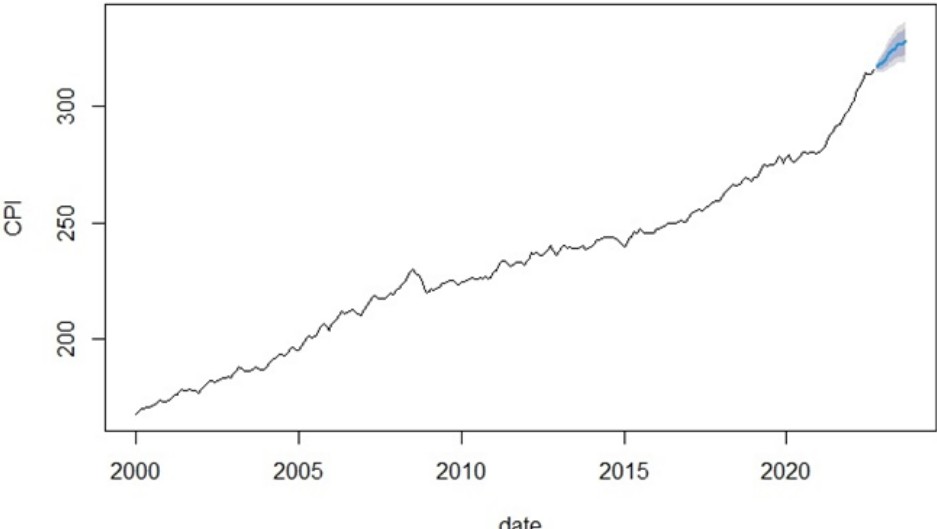

**Figure 9.** SARIMA(0,1,1)(0,0,2) prediction of CPI development in Los Angeles. Own processing based on data from the US Bureau of Labor Statistics.

Even in Los Angeles, growth in consumer prices could be predicted for most of the next calendar year. The inflation jump was the highest when compared to the other indices, as the difference between the beginning and the end of the prediction was more than USD11. According to the predictive model, at the beginning of Q4 of 2023, Los Angeles will show the highest inflation rate of all the US agglomerations under review. The optimal SARIMA model setting was (0, 1, 1)(0, 2, 2).

To obtain a better picture of the relevance of the statistical methods (or rather the predictions provided by the SARIMA model), Appendix B shows the graphs of residuals and lags within the ACF (auto-correlation function), specifically Figures A1–A4. The statistical apparatus can be perceived as useful since seasonality did not have any extreme effect on the values, which means there were no significant fluctuations in lags beyond the given limits.

## 5. Discussion

The goal of this paper was to determine the potential dependence between the price of coffee and the inflation rate. For a more detailed analysis, two research questions were formulated:

RQ1: How strong is the correlation between the coffee price in ICE Futures US and the level of inflation in selected US agglomerations?

The application of the Pearson correlation coefficient confirmed the existence of a strong direct dependence. Based on the findings, the strongest correlation between coffee price and inflation was identified in the agglomerations of New York and Chicago (with the resulting r value of 0.61). A slightly weaker dependence was identified in the Los Angeles agglomeration (resulting r value of 0.57). It is thus possible to confirm the statements of Salisu et al. [12], whose findings were similar. The results obtained are also in line with Halka and Kotlowski [25] and their findings concerning the existence of the correlation between the shocks in commodity exchange and the level of inflation. A similar conclusion was made by Forbes [26].

RQ2: What trend can be expected for the price of coffee and inflation?

Using the SARIMA model, significant growth was predicted for all CPIs where the inflation growth increased the value of the consumer basket by nearly USD10 in New York and Chicago (this refers to the difference between the first and last months of the predicted period, i.e., October 2022 and September 2023). In Los Angeles, this difference amounted to more than USD11.

The predicted coffee price shows a rather fluctuating trend. The market price of coffee is expected to reach a new level at which it will fluctuate. In this context, the future validity

of RQ1 is limited since the strength of the correlation will be low and changeable. To some extent, it is possible to compare the prediction with the findings of Vochozka et al. [16], who defined coffee as an inferior and non-necessary good based on income and price elasticities. Inflationary growth may not have such a fundamental effect on demand.

In line with the goal of this paper, it is possible, with caution, to state a stronger direct correlation between coffee price and inflation in the market of three defined US agglomerations (New York, Chicago, Los Angeles) based on the time series of more than 22 years. A prediction, whose importance is mentioned by Vochozka et al. [14], rather disrupts the essence of the correlation in its future concept.

The limitation of this research is the specification of the market. There is room for analyzing the relationship between coffee prices and inflation in other markets. Further research could focus on the causality between coffee prices and inflation, as the identified correlation as such is not a sufficient explanation of how the economic variables interact with each other. It could be inflation, not coffee price, that is the cause of the shocks in the commodity exchange. Besides coffee, there are other commodities that should be examined in terms of their relationship to inflation (given the result of the prediction model). Furthermore, the application of a predictive statistical apparatus instead of the seasonal ARIMA (or SARIMA) model, or rather, the manual version of SARIMA instead of the automated version, should be considered. The principle of linear autoregression used by the ARIMA models is also a limiting factor; models with non-linear regressions should be applied instead (such as Artificial Neural Networks) or a combination of these models for more accurate predictions.

This paper may be beneficial for the creators of fiscal and monetary policies. The relationship between coffee price and CPIs is significant and can thus be helpful. The results could be particularly useful in analyses focused on the function of commodity markets as well as in statistical studies that are the responsibility of statistical offices. Moreover, the results could also be applied within coffee supply-customer chains, especially for traders with considerable bargaining power in the market. The consumer benefit of the results can be seen in the possibilities of setting future rational expectations, which can also be said in the case of producers.

## 6. Conclusions

The goal of this paper was to determine the relationship between the price of coffee traded on the exchange market and the level of inflation in three significant US agglomerations—New York, Chicago, and Los Angeles—and, subsequently, predict future development. Based on the historical data, or time series of more than 22 years with a monthly data frequency, a very close correlation could be confirmed between coffee price and CPIs of the given US agglomerations using the classic Pearson correlation coefficient. A 12-month SARIMA prediction from October 2022 to September 2023 identified the growing trend in CPIs, but coffee prices fluctuated around a new price level. In such a situation, the direct correlation would be disrupted.

Given the above, the goal of this paper can be considered achieved. The major limitation of this research was the narrow specification of the market. Here, there is room for new territorial definitions of the market. The autoSARIMA model was also a limitation; in the future, this model should be compared with other predictive models, or even the predictions of selected predictive models should be mutually compared within one paper.

Determining the causality and resulting designation affecting economic variables in the relationship between coffee price and inflation rate represents another research opportunity. It is also possible to observe other significant traded commodities in the context of inflation rates and the findings of this research and other research opportunities.

These findings can be useful in setting investment strategies on a technical basis. The results are also applicable to the creators of monetary and fiscal policies since they may provide valuable information on the market and macroeconomic situation.

**Author Contributions:** Conceptualization, M.V. and Z.R.; methodology, S.J.; software, S.J.; validation, M.V., S.J. and Z.R.; formal analysis, Z.R.; investigation, S.J.; resources, S.J.; data curation, M.V.; writing—original draft preparation, S.J.; writing—review and editing, Z.R.; visualization, S.J.; supervision, M.V. All authors have read and agreed to the published version of the manuscript.

**Funding:** This research received no external funding.

**Institutional Review Board Statement:** Not applicable.

**Informed Consent Statement:** Not applicable.

**Data Availability Statement:** The datasets used in this contribution are sourced from: https://data.bls.gov/cgi-bin/surveymost and https://finance.yahoo.com/quote/KC=F?p=KC=F&.tsrc=fin-srch. The data were accessed on 15 November 2022.

**Conflicts of Interest:** The authors declare no conflict of interest.

## Appendix A

**Table A1.** Prediction of the development of coffee price in the commodity exchange ICE Futures US. Own processing.

| Date | Point Forecast | Lo 80 | Hi 80 | Lo 95 | Hi 95 |
|---|---|---|---|---|---|
| Oct 2022 | 226.8081 | 211.6980 | 241.9181 | 203.6992 | 249.9169 |
| Nov 2022 | 220.3462 | 201.0933 | 239.5990 | 190.9014 | 249.7909 |
| Dec 2022 | 222.0414 | 197.7798 | 246.3030 | 184.9364 | 259.1464 |
| Jan 2023 | 220.7782 | 192.1868 | 249.3697 | 177.0514 | 264.5051 |
| Feb 2023 | 220.7799 | 187.9747 | 253.5851 | 170.6087 | 270.9511 |
| Mar 2023 | 222.0933 | 185.3221 | 258.8645 | 165.8566 | 278.3300 |
| Apr 2023 | 222.4028 | 181.8441 | 262.9615 | 160.3736 | 284.4320 |
| May 2023 | 220.8393 | 176.6722 | 265.0065 | 153.2916 | 288.3871 |
| Jun 2023 | 220.5905 | 172.9749 | 268.2061 | 147.7688 | 293.4123 |
| Jul 2023 | 222.4287 | 171.5135 | 273.3438 | 144.5607 | 300.2966 |
| Aug 2023 | 219.1673 | 165.0893 | 273.2452 | 136.4622 | 301.8724 |
| Sep 2023 | 221.6133 | 164.4986 | 278.7280 | 134.2638 | 308.9627 |

**Table A2.** Predicted CPI trend in New York. Own processing.

| Date | Point Forecast | Lo 80 | Hi 80 | Lo 95 | Hi 95 |
|---|---|---|---|---|---|
| Oct 2022 | 314.2643 | 313.1817 | 315.3469 | 312.6086 | 315.9200 |
| Nov 2022 | 314.6352 | 312.9129 | 316.3575 | 312.0012 | 317.2692 |
| Dec 2022 | 314.9558 | 312.7360 | 317.1755 | 311.5610 | 318.3506 |
| Jan 2023 | 316.3744 | 313.7421 | 319.0067 | 312.3487 | 320.4002 |
| Feb 2023 | 317.1168 | 314.1267 | 320.1068 | 312.5438 | 321.6897 |
| Mar 2023 | 318.6589 | 315.3492 | 321.9686 | 313.5971 | 323.7207 |
| Apr 2023 | 320.1009 | 316.4997 | 323.7021 | 314.5933 | 325.6084 |
| May 2023 | 321.0848 | 317.2140 | 324.9556 | 315.1649 | 327.0046 |
| Jun 2023 | 323.1373 | 319.0145 | 327.2601 | 316.8320 | 329.4426 |
| Jul 2023 | 323.0866 | 318.7263 | 327.4469 | 316.4181 | 329.7551 |
| Aug 2023 | 323.6243 | 319.0388 | 328.2098 | 316.6114 | 330.6372 |
| Sep 2023 | 324.3857 | 319.5856 | 329.1858 | 317.0446 | 331.7269 |

**Table A3.** Predicted CPI trend in Chicago. Own processing.

| Date | Point Forecast | Lo 80 | Hi 80 | Lo 95 | Hi 95 |
|------|---------------|-------|-------|-------|-------|
| Oct 2022 | 279.7314 | 278.4290 | 281.0338 | 277.7396 | 281.7232 |
| Nov 2022 | 279.7646 | 277.7111 | 281.8180 | 276.6241 | 282.9051 |
| Dec 2022 | 280.0849 | 277.3842 | 282.7855 | 275.9546 | 284.2151 |
| Jan 2023 | 281.3120 | 278.0378 | 284.5862 | 276.3045 | 286.3195 |
| Feb 2023 | 282.3707 | 278.5811 | 286.1602 | 276.5751 | 288.1663 |
| Mar 2023 | 283.7852 | 279.5275 | 288.0429 | 277.2736 | 290.2968 |
| Apr 2023 | 285.0122 | 280.3249 | 289.6996 | 277.8436 | 292.1809 |
| May 2023 | 286.9087 | 281.8236 | 291.9938 | 279.1317 | 294.6856 |
| Jun 2023 | 288.3683 | 282.9120 | 293.8246 | 280.0236 | 296.7130 |
| Jul 2023 | 288.8076 | 283.0025 | 294.6127 | 279.9294 | 297.6858 |
| Aug 2023 | 289.1165 | 282.9816 | 295.2514 | 279.7340 | 298.4991 |
| Sep 2023 | 289.7102 | 283.2620 | 296.1584 | 279.8485 | 299.5719 |

**Table A4.** Predicted CPI trend in Los Angeles. Own processing.

| Date | Point Forecast | Lo 80 | Hi 80 | Lo 95 | Hi 95 |
|------|---------------|-------|-------|-------|-------|
| Oct 2022 | 316.4195 | 315.1146 | 317.7245 | 314.4238 | 318.4152 |
| Nov 2022 | 317.4169 | 315.2666 | 319.5671 | 314.1284 | 320.7053 |
| Dec 2022 | 318.2139 | 315.4672 | 320.9605 | 314.0132 | 322.4145 |
| Jan 2023 | 319.5153 | 316.2804 | 322.7502 | 314.5679 | 324.4627 |
| Feb 2023 | 320.2149 | 316.5563 | 323.8736 | 314.6196 | 325.8103 |
| Mar 2023 | 322.1882 | 318.1501 | 326.2263 | 316.0125 | 328.3639 |
| Apr 2023 | 323.3268 | 318.9419 | 327.7116 | 316.6207 | 330.0328 |
| May 2023 | 324.4784 | 319.7723 | 329.1845 | 317.2810 | 331.6758 |
| Jun 2023 | 325.9082 | 320.9014 | 330.9150 | 318.2510 | 333.5655 |
| Jul 2023 | 326.0954 | 320.8050 | 331.3859 | 318.0044 | 334.1865 |
| Aug 2023 | 326.5591 | 320.9994 | 332.1187 | 318.0564 | 335.0618 |
| Sep 2023 | 327.4955 | 321.6792 | 333.3119 | 318.6002 | 336.3909 |

**Appendix B**

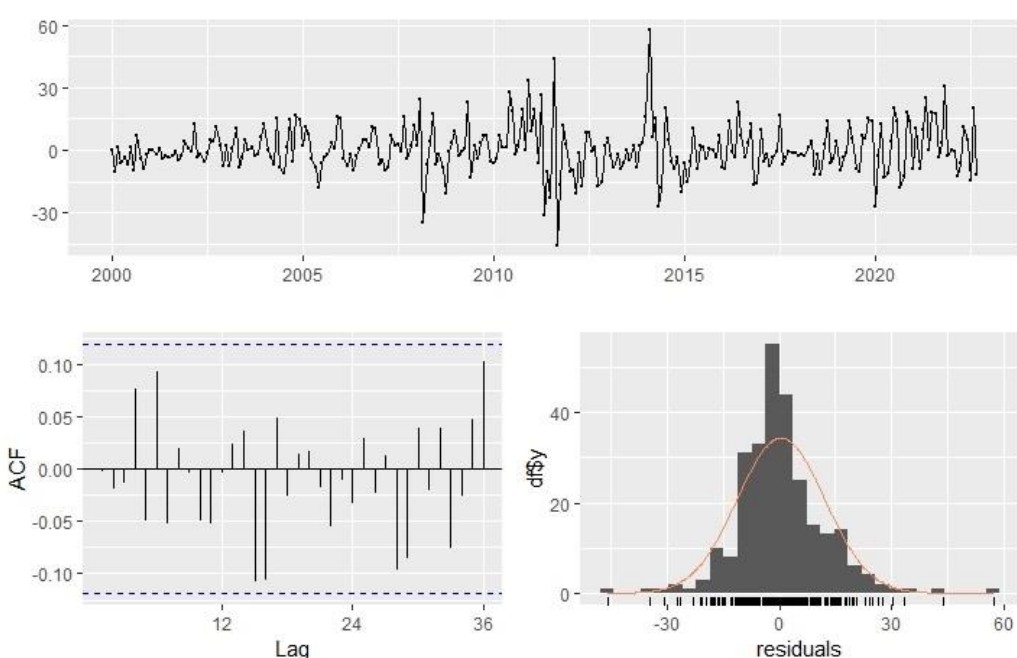

**Figure A1.** Residuals, ACF, and lags in prediction of Coffee. Own processing.

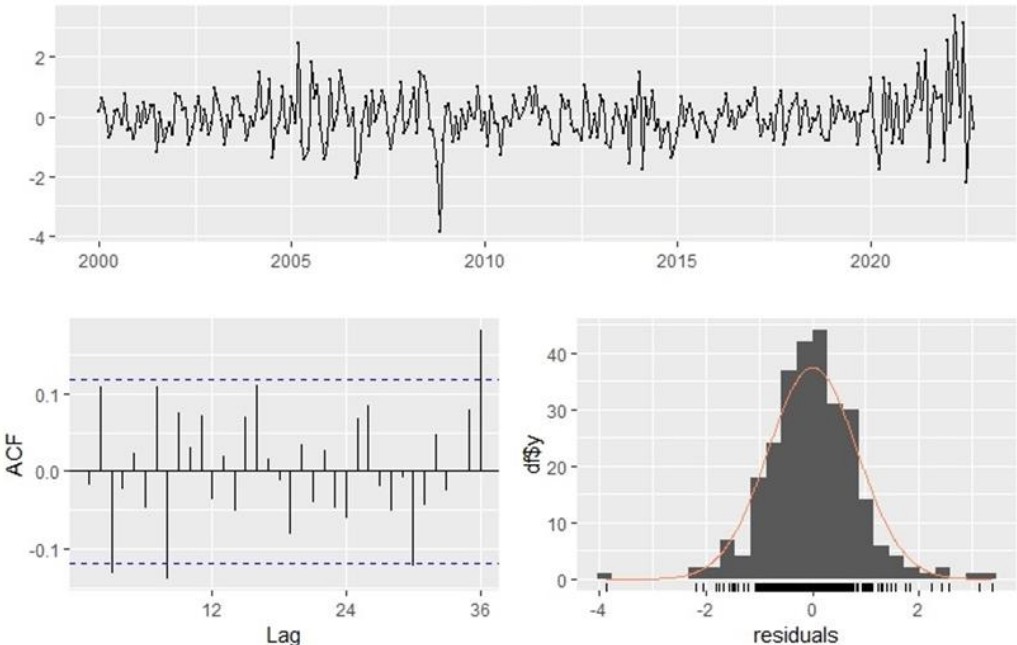

**Figure A2.** Residuals, ACF, and lags in prediction of CPI New York. Own processing.

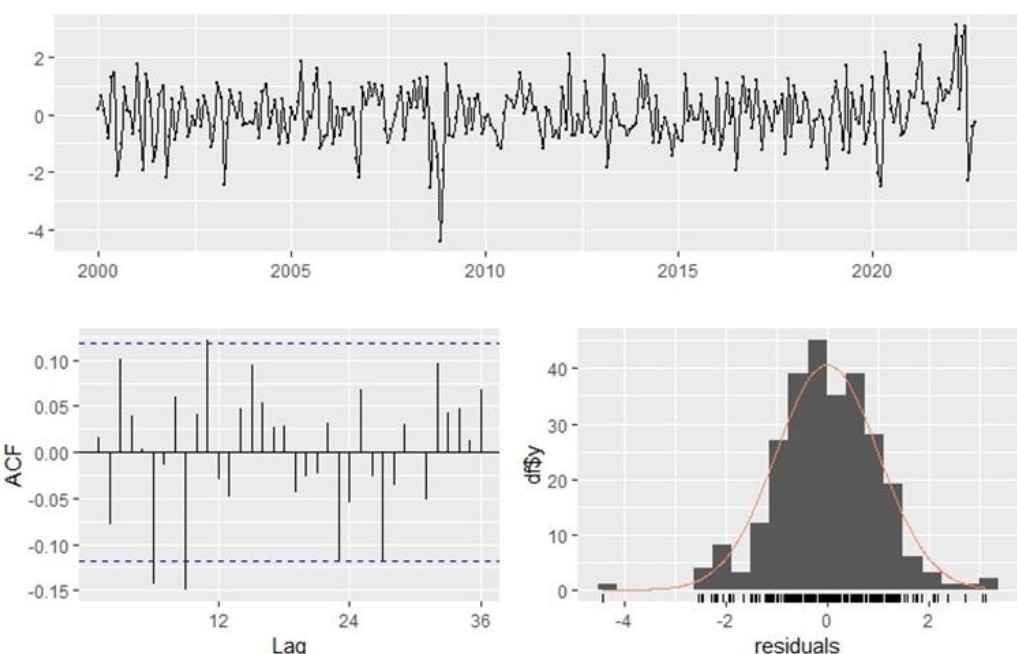

**Figure A3.** Residuals, ACF, and lags in prediction of CPI Chicago. Own processing.

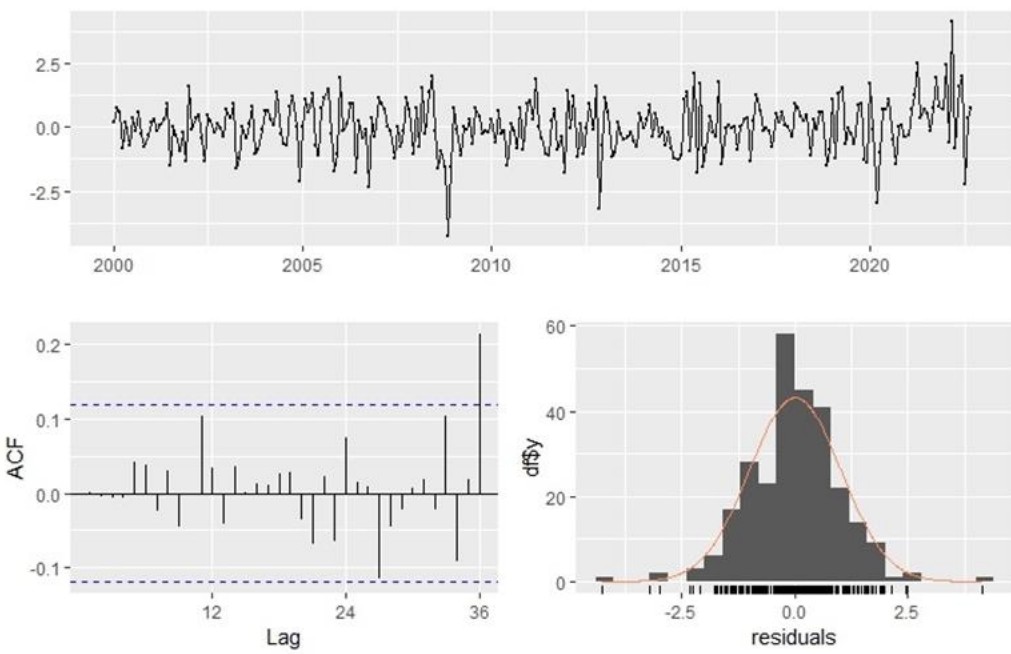

**Figure A4.** Residuals, ACF, and lags in prediction of CPI Los Angeles. Own processing.

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
