# Peer review of "Coffee as an Identifier of Inflation in Selected US Agglomerations"

_forecasting, doi:10.3390/forecast5010007_

Round 1

Reviewer 1 Report

The paper titled “ Coffee as an identifier of inflation in selected US agglomera-2 tions?” addresses a potentially interesting topic .I have the following comments:

-         What is novelty in this study? Just emphasizing on the applied method can not be a novelty. You have to work more on the introduction and define clear objectives that are linked to the introduction.

-          

-         Equation 1, (Pearson 153 correlation coefficient) can be removed from the paper as it is available in many textbooks.

-         What is the relevance of your study in general for consumers and producers?

-         The quality of forecasted model by ARIMA is not mentioned in the text. In the Did you have the valuses of root mean square, mean absolute error.

-        The discussion should connect the introduction by way of the research questions or hypotheses you posed and the literature you reviewed. Hence, it should be some comparison of your results with literature.

I recommend the publication of this paper with major revision.

Author Response

Dear reviewer,

We are responding to your comments. Changed or supplemented is indicated in red color. There was a clearer definition of the objective and the connection with the introduction. The Pearson R equation is removed, the relevance of the applied SARIMA model is added to the methodology.

We believe that the text now meets all requirements.

Yours Sincerely,

Marek Vochozka, Svatopluk Janek, Zuzana Rowland

Reviewer 2 Report

1) The research problem indirectly tackled with this study is similar to "lipstick index" and other claimed-to-be trackers of recession. There were many such ideas to use one simple index to track some other more complicated factor. Nothing about such things is cited in the submitted manuscript.

2) Figure 1 is not enough to discuss long-term. Cointegration analysis should be done with some test(s).

3) Quite small sample is considered. Coffee price should be avaialble in monthly periods for a much longer time.

4) In this context, time-varying correlations could be interesting to check for robustness of the results.

5) Figures 4 and 5 clearly show some non-linear relationship, but linear regression is postulated. Except or instead non-linearity maybe there are some sub-gropus? Anyways, I have some objections towards assumption of linearity from looking on these figures.

6) In case of forecasting in-sample and out-of-sample periods are unknown. Besides, the out-of-sample period which I presumed from analysis is very very short. It clearly impacts the outcomes.

7) How ARIMA lags were chosen?

8) There is no benchmark forecasting model.

9) Which prices (adjusted, closing?) were used?

10) Actually, all forecasting in true out-of-sample and therefore there is no evaluation of the proposed model over some period back.

Most of my concerns about lack of cointegration tests, non-linearity of sub-gropus, time-varying effects and lack of benchmark, etc. make me suggest that the paper must be significantly improved in order to be meaningful. Anyways, the idea of the study is interesting and worth publication if improved.

Author Response

Dear reviewer,

We are responding to your comments. Changed or supplemented is indicated in red color. Figure 1 has a clearly informative function. Figures 3 and 4 illustrate a linear relationship (all coded in R). The auto ARIMA model automatically selects the most appropriate correlations between time series and predictions, this also applies to lags. Time series is optimal - our experience is not good with longer data. Distortion occurs very often. Comparison with other predictors is lacking, however at the ARIMA model level there was testing between the automatic model and the manual model. The results were compared and evaluated as equal.

We believe that the text now meets all requirements.

Yours Sincerely,

Marek Vochozka, Svatopluk Janek, Zuzana Rowland

Round 2

Reviewer 1 Report

I checked the revised manuscript titled “ Coffee as an identifier of inflation in selected US agglomera-2 tions?”

 The authors revised the manuscript according to my comments.

 Hence, I recommend the publication of this manuscript.

Reviewer 2 Report

Non of my previous comments were properly answered. Please provide a response in the following manner:

Answer to Question 1): ...

...

Answer to Question 10): ...

In my opinion the paper is still needing significant improvements.

Round 3

Reviewer 2 Report

1) "Our experience evaluates the data as sufficient. Too large data series (over 30 years) are very often distorted the model results (especially in monthly and daily frequency)." This is ridiculous (sic!) answer. You know that taking larger data set would invalidate your outcomes so you take smaller sample (less information) with purspose(!) to claim some conclusions. Of course, all your conclusions are then invalid... Such an approach and argument in my opinion totally disqualifies the paper as the underlying motivation and "scientific" approach is wrong.

2) "The figures clearly illustrate a linear relationship." I am not going to dispute that what is clearly see it not what I see. That white is black and black is white. Please, compute MSE of your "surely" linear regression and compare with the modelled values. You will see how enourmous big errors your will obtain meaning the fittnes of the model is poor.

3) There is still no info on in-sample and out-of-sample periods.

4) There must be some benchmark model(s). https://robjhyndman.com/hyndsight/benchmarks/

This paper is not suitable for a good journal, especially connected with forecasting, statistics or econometrics. It contains serious methodoogical flaws and the answers from authors indicate that they have not performed proper statistical analysis, neither they are willing to make such. The above point are a total must before considering publication of this paper in my opinion.